# Static Compressive Properties of Polypropylene Fiber Foam Concrete with Concave Hexagonal Unit Cell

**Zhiqiang Yin** [1,2,*], **Zhenguo Shao** [1], **Chao Qi** [1], **Haoyuan Wu** [1], **Jianen Wang** [1] and **Lulu Gao** [1]

1    School of Mining Engineering, Anhui University of Science and Technology, Huainan 232001, China
2    Coal Mine Safety Mining Equipment Innovation Center of Anhui Province, Huainan 232001, China
*    Correspondence: zhqyin@aust.edu.cn; Tel.: +86-183-5548-6807

**Abstract:** For the purpose of studying the influence of fiber on the negative Poisson's ratio effect of foam concrete, a concave hexagonal unit cell structure of polypropylene fiber foam concrete was proposed. The effects of different fiber volume contents on the structural mechanical parameters, Poisson's ratio, and energy absorption capacity of the unit cells were studied by static compression of concave hexagonal unit cells and cube specimens. The results show that the compressive strength of foam concrete is reduced by adding polypropylene fiber, and the peak stress of concave hexagonal unit cells decreases less rapidly than that of cube specimens. The proper amount of polypropylene fiber can enhance the deformation ability of the unit cells in foam concrete, and the Poisson's ratio of the unit cells in foam concrete with 1.5% fiber content is the lowest. In the process of failure of concave hexagonal unit cells, the failure phenomenon is mainly concentrated on the concave surfaces on both sides, and the cracks are distributed in the form of "upper left and lower right" or "lower left and upper right". When the content of polypropylene fiber is 0.5%, the total energy absorbed by the concave hexagonal cells in the compression deformation process increases by 12.98%.

**Keywords:** foam concrete; fiber; negative Poisson's ratio; concave hexagon; energy absorption

## 1. Introduction

It is generally believed that almost all materials have positive Poisson's ratios, that is, these materials shrink laterally when they are stretched. The negative Poisson's ratio effect refers to the lateral expansion of the material in the elastic range when it is stretched, and the transverse direction of the material shrinks instead when compressed. In general, the negative Poisson's ratio effect is not commonly observed in materials, but some materials with special structures have been found to have a negative Poisson's ratio effect in recent years. Materials with negative Poisson's ratios have improved shear modulus, seismic absorption capacity, fracture resistance, and indentation resistance compared with ordinary structural materials [1–3]. The discovery of negative Poisson's ratio materials provided a new idea and method for studying materials and structures with special mechanical properties [4,5]. The typical types of negative Poisson's ratio cytosol are mainly concave polygonal structures, chiral structures, rotating rigid structures, and other structures; the most studied ones are concave hexagonal structures [6–8]. As a new material, negative Poisson's ratio material can be used as both structural and functional material. The design of negative Poisson's ratio structures can be obtained by designing the concave angle structure to show the negative Poisson's ratio effect in addition to the direct preparation of negative Poisson's ratio material [9–11].

Because of the special properties of negative Poisson's ratio materials with tensile expansion, they are widely used in the civil engineering and medical fields. In recent years, Zhang Junbao et al. [12], Tao Zhigang et al. [13], and He Manchao et al. [14] studied the mechanical properties of negative Poisson's ratio anchor cables and their applications in civil engineering. Wang Junyan et al. [15] investigated the influence law of various

parameters on the adhesion performance between negative Poisson's reinforcement and ultra-high performance concrete, and the test showed that the adhesion performance of high strain-reinforced ultra-high performance concrete and negative Poisson's reinforcement was better than that of the low version. The strain-reinforced and strain-softened types were found to be more effective than the low-strain-reinforced type. In the medical field, LIN Cheng et al. [16] used 4D printing to develop a shape-memory personalized vascular scaffold with a negative Poisson's ratio structure, which was shown to be promising for the treatment of vascular stenosis through in vitro tests. Huang Weifeng et al. [17] designed a vascular stent as a six-ligament chiral honeycomb structure to improve the matching of the vascular stent, in order to intimal deformation and reduce the probability of intimal hyperplasia. Meanwhile, the application of negative Poisson's ratio structures in explosion-proof cushioning materials is also worthy of attention, due to its excellent energy absorption properties. Zhu Jiansheng et al. [18] investigated the dynamic response of negative Poisson's ratio structured load-reducing components under an artillery firing load, which provided a reference for the design of bullet-loaded devices against high overload. Zhao Yuwei et al. [19] used a double-arrow honeycomb material with a negative Poisson's ratio effect as the core part of a sandwich core structure. The application of the double-arrow honeycomb material with a negative Poisson's ratio effect as the core part of a sandwich structure was applied to a lower limb protection device for military vehicle occupants. Zhao Ying et al. [20] proposed a new, negative Poisson's ratio, non-pneumatic tire support body structure to improve the stress level stiffness characteristics and lateral stability of non-pneumatic tires.

The negative Poisson's ratio structure and the influence of material property parameters on the energy absorption effect of composite structure have been considered comprehensively, and the negative Poisson's ratio structure has been combined with the design of composite energy-absorbing materials. For example, Weitao Lv et al. [21] modeled a sandwich panel consisting of two metal panels and a three-dimensional isotropic foam core with a negative Poisson's ratio, and the one with the smallest Poisson's ratio had the highest blast resistance, as shown through finite element simulations. Yao Zhang et al. [22] used sandwich composite specimens prepared from 3D printed dot matrix cores and glass/epoxy composite sheets. They performed low-velocity impact tests, and found that the location of the impact point affected the impact mechanical response and energy absorption capacity of the sandwich panels. Zhao Changfang et al. [23] proposed a combined mold of laminated carbon fiber-reinforced plastic negative Poisson's ratio unit structure specimens, which were made by the high-temperature hot pressing process, and experimentally verified the potential engineering application of the CFRP + NPRS series in lightweight vibration damping and energy absorption. Zhou Hongyuan et al. [24] put forward that a negative Poisson's ratio filled structure can be obtained by filling the negative Poisson's ratio structure with foam concrete with different densities, and the test showed that an appropriate amount of foam concrete filling significantly improved the energy absorption capacity of the negative Poisson's ratio structure. Ma Yanxuan et al. [25] introduced the negative Poisson's ratio structure into concrete for the purpose of improving the energy storage modulus and the blast absorption capacity of concrete.

This paper studies the influence of polypropylene fiber content on the peak stress, failure mode, deformation capacity, and energy absorption capacity of foam concrete with concave hexagonal unit cells. The concave hexagonal unit cells were obtained by pouring foam concrete into self-made molds. Static compression tests were carried out on concave hexagonal unit cells and cube specimens. Herein, the Poisson's ratio, failure mode, and energy absorption capacity of different concave hexagonal unit cells are compared, and the energy absorption principle of concave hexagonal unit cells in fiber foam concrete is discussed.

## 2. Materials and Methods

### 2.1. Raw Material and Proportioning

The materials used in this test are mainly divided into cementitious materials, water, fiber, foaming agent, and mold materials. Cementitious materials were PO.42.5 ordinary silicate cement and Class I coal ash. Water was local tap water from Huainan. LG-2258 cement foaming agent was used as the foaming agent. Polypropylene fiber is a common concrete additive, which can improve the deformation capacity of concrete [26]. Thus, polypropylene fiber with a length of 9 mm was selected. Mold materials were high-density foam board and foam special glue.

According to 0%, 0.5%, 1.0%, 1.5%, 2.0%, and 2.5% fiber volume doping, a total of six groups of 50 mm × 50 mm × 50 mm fiber foam condensate negative Poisson's ratio structure specimens were set up in the experiment. In addition, cubic specimens with side lengths of 50 mm, in the same proportion, were set up as the comparison groups. The water–binder ratio was set to 0.4, and the water content of the foam agent was not included in it [27]. The ingredients and proportions are shown in Table 1.

**Table 1.** The proportions of fiber foam concrete.

| Fiber Content | Concrete/(kg·m$^{-3}$) | Coal Ash/(kg·m$^{-3}$) | Water/(kg·m$^{-3}$) | Foam/(kg·m$^{-3}$) | Polypropylene Fiber/(kg·m$^{-3}$) |
|---|---|---|---|---|---|
| 0% | 600 | 400 | 400 | 7.293 | 0 |
| 0.50% | 600 | 400 | 400 | 7.293 | 4.550 |
| 1.00% | 600 | 400 | 400 | 7.293 | 9.100 |
| 1.50% | 600 | 400 | 400 | 7.293 | 13.650 |
| 2.00% | 600 | 400 | 400 | 7.293 | 18.200 |
| 2.50% | 600 | 400 | 400 | 7.293 | 22.750 |

### 2.2. Specimen Design and Fabrication

The concave hexagon is one of the typical structures of negative Poisson's ratio, which is simple in shape and convenient for carrying out mechanical ballast and impact tests. Therefore, the concave hexagon unit cell was used as the structural design of foam concrete negative Poisson structure specimens. The section shape and size of the specimen for the quasi-static compression test are shown in Figure 1.

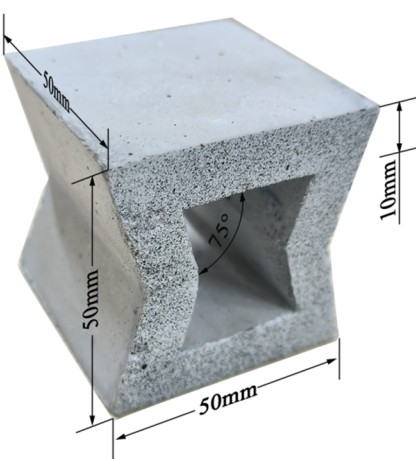

**Figure 1.** Design of the concave hexagonal unit cell of foam concrete.

The size of the concave hexagonal cell was set to 50 mm × 50 mm × 50 mm, the cell wall thickness was set as 10 mm, and the internal concave angle was set as 75°. To create the concave hexagonal unit cells, the mold was firstly designed and created, and then it was cut and assembled with high-strength foam. The high-density foam sheets were cut

by the foam cutting machine according to the size requirements, and the foam parts were checked and corrected, then assembled with foam special adhesive. The molds were left alone for 48 h after assembly, as they could be used for pouring the test specimen only after the glue was completely consolidated.

Polypropylene fiber foam concrete was mixed according to the proportion table and poured into the mold. All of the poured specimens were allowed to stand for 48 h before demolding. Since the molds were spliced by foam plates, the foam could be directly removed with tools to obtain concave hexagonal unit cells. After demolding, all specimens were immersed in a container containing a calcium hydroxide-saturated solution for 28 days, and then cured in a laboratory at a temperature of 20 ± 2 °C.

### 2.3. Static Compressive Testing

In the test, six groups of concave hexagonal unit cells and six groups of cube specimens with different fiber contents were loaded on the RMT testing machine, with three samples for each group. The samples were marked: "A" is the concave hexagon unit cell, and "F" is the cube specimen; the middle number represents the fiber volume content; and the last number represents the sample number. After completing the size check and weighing, the specimens were placed in the RMT testing machine for static compression. When the sample deformed rapidly, it was deemed to be damaged. The peak stress and peak strain of the sample were recorded and used to calculate the average value of each group. The stress-strain data recorded by the RMT testing machine were derived, and the stress-strain curves of concave hexagonal cells with different fiber contents were sorted out. The energy absorption efficiency was evaluated by using the formulas.

### 2.4. Digital Speckle Correlation Method

The deformation of concave hexagonal cells during compression was detected by the digital speckle correlation method. Paint was sprayed onto the surface of the concave hexagonal cell to form a uniform and dense speckle field. A digital camera was used to record the compression process of the sample. The image acquisition equipment and processing schematic are shown in Figure 2. The compression process was recorded as a video, and screenshots were taken at 1 s intervals. The optical experiment software GOM Correlate was used to process the video screenshot in order to obtain the displacement program of the concave hexagonal unit cells, and the change value of the transverse distance of the midpoint of the concave on both sides of the specimens was measured to obtain the transverse deformation. Combined with the longitudinal deformation measured by the RMT testing machine in the ballast experiment, the macro Poisson's ratio of concave hexagonal unit cells was calculated through the definition formula of Poisson's ratio.

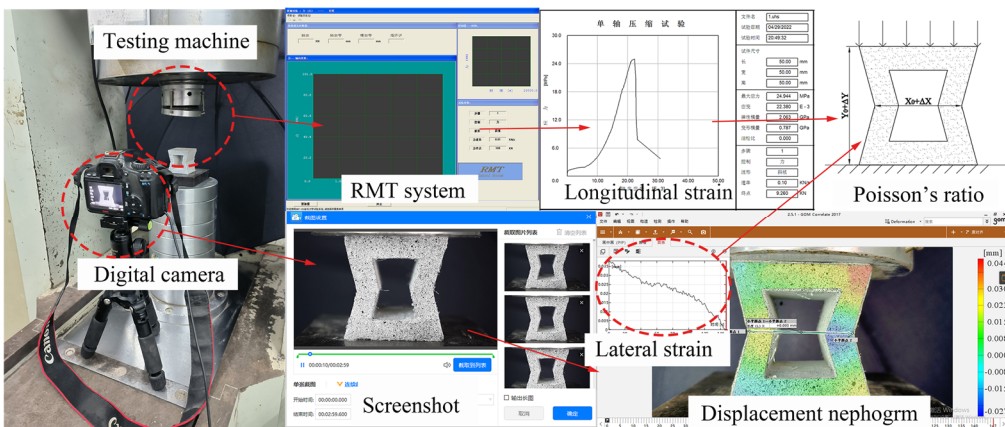

**Figure 2.** Experimental procedure for DSCM of the concave hexagonal unit cell.

## 3. Results and Discussion

### 3.1. Mechanical Property

By static compression test, the peak stress and peak strain of the specimens were obtained and the average value of each group of samples was taken, and the data parameters were obtained, as shown in Table 2.

**Table 2.** Mechanical parameters of specimens.

| Sample | Stress/MPa | Average/MPa | Strain/$10^{-3}$ | Average/$10^{-3}$ |
|---|---|---|---|---|
| A-0-1 | 9.333 | | 7.260 | |
| A-0-2 | 8.551 | 8.919 | 7.450 | 7.303 |
| A-0-3 | 8.872 | | 7.200 | |
| A-0.5-1 | 9.296 | | 7.700 | |
| A-0.5-2 | 8.816 | 8.843 | 7.780 | 7.547 |
| A-0.5-3 | 8.416 | | 7.160 | |
| A-1.0-1 | 8.852 | | 8.020 | |
| A-1.0-2 | 8.591 | 8.667 | 7.660 | 7.737 |
| A-1.0-3 | 8.559 | | 7.530 | |
| A-1.5-1 | 8.036 | | 7.90 0 | |
| A-1.5-2 | 7.822 | 7.911 | 7.880 | 7.793 |
| A-1.5-3 | 7.874 | | 7.600 | |
| A-2.0-1 | 7.072 | | 7.900 | |
| A-2.0-2 | 6.950 | 7.150 | 7.360 | 7.590 |
| A-2.0-3 | 7.427 | | 7.510 | |
| A-2.5-1 | 6.591 | | 7.250 | |
| A-2.5-2 | 6.416 | 6.573 | 7.480 | 7.380 |
| A-2.5-3 | 6.712 | | 7.410 | |
| F-0-1 | 47.768 | | 15.180 | |
| F-0-2 | 44.312 | 46.839 | 16.140 | 15.813 |
| F-0-3 | 48.436 | | 16.120 | |
| F-0.5-1 | 47.768 | | 17.80 | |
| F-0.5-2 | 46.368 | 46.741 | 15.380 | 16.647 |
| F-0.5-3 | 46.088 | | 16.760 | |
| F-1.0-1 | 43.064 | | 17.460 | |
| F-1.0-2 | 44.448 | 44.139 | 17.760 | 17.687 |
| F-1.0-3 | 44.906 | | 17.840 | |
| F-1.5-1 | 37.872 | | 16.980 | |
| F-1.5-2 | 40.552 | 39.208 | 17.840 | 17.980 |
| F-1.5-3 | 39.200 | | 19.120 | |
| F-2.0-1 | 37.744 | | 16.060 | |
| F-2.0-2 | 33.808 | 34.448 | 16.900 | 17.287 |
| F-2.0-3 | 31.792 | | 18.900 | |
| F-2.5-1 | 28.160 | | 16.460 | |
| F-2.5-2 | 27.728 | 27.525 | 16.700 | 16.240 |
| F-2.5-3 | 26.688 | | 15.560 | |

Figure 3 shows the trend of the effect of different fiber contents on the peak stress of cubic specimens and the concave hexagonal unit cells. It can be seen that the variation of mechanical parameters in static compression for foam concrete with different fiber contents in the concave hexagonal unit cells has similar characteristics to that of the cubic specimens.

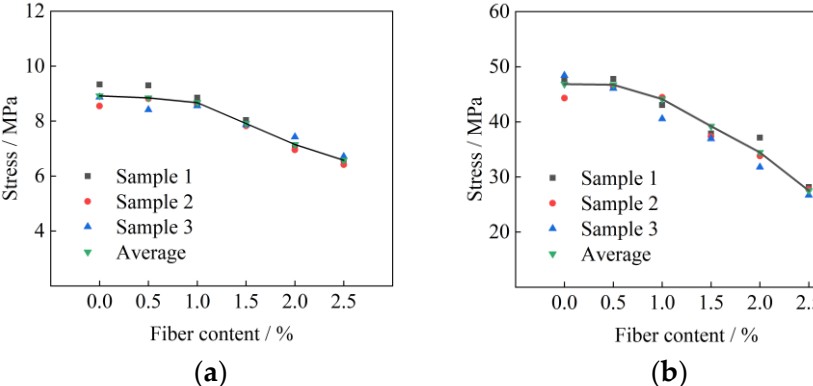

**Figure 3.** Effect of different fiber content on the peak stress of the specimen: (**a**) concave hexagonal unit cells; (**b**) cubic specimens.

As seen in Figure 3, with the increase in polypropylene fiber content, the peak stress of concave hexagonal cells and cubics decreases gradually. When the volume fraction of polypropylene fiber increased from 0% to 0.5%, the average peak stress of the concave hexagonal unit cell decreased by 0.85%, and the average peak stress of the cube decreased by 0.21%. When the volume fraction of polypropylene fiber increased from 0.5% to 1.0%, the average peak stress of concave hexagonal unit cell decreased by 1.99%, and the average peak stress of cube decreased by 5.57%. With the further increase in polypropylene fiber, the peak stress of foam concrete weakened by polypropylene fiber was more significant. When the volume fraction of polypropylene fiber increased from 1.0% to 1.5%, the average peak stress of the concave hexagonal unit cell decreased by 8.72%, and the average peak stress of the cube decreased by 11.17%. When the volume fraction of polypropylene fiber increased from 1.5% to 2.0%, the average peak stress of concave hexagonal unit cell decreased by 9.61%, and the average peak stress of the cube decreased by 12.14%. When the volume fraction of polypropylene fiber increased from 1.5% to 2.0%, the average peak stress of concave hexagonal unit cell structure decreased by 8.07%, and the average peak stress of cube decreased by 20.10%.

It can be seen that the reduction in peak stress of the concave hexagonal unit cell by polypropylene fiber content is smaller than that of the cube. In order to explore the influence of cell structures on their pick stress, a two-way ANOVA was conducted on the peak stress of two kinds of specimens. The processing results are shown in Table 3. The *p*-value of the structure is less than 0.050, and it more clearly verifies that there are differences in the change in peak stress of cells with different structures.

**Table 3.** Results of the two-way ANOVA.

| Source | Class III Sum of Squares | Freedom | Mean Square | F | *p*-Value |
|---|---|---|---|---|---|
| Revised Model | 3222.721a | 6 | 537.120 | 23.732 | 0.002 |
| Intercept | 6862.314 | 1 | 6862.314 | 303.208 | 0.000 |
| Fiber content | 187.825 | 5 | 37.565 | 1.660 | 0.296 |
| Structure | 3034.897 | 1 | 3034.897 | 134.096 | 0.000 |
| Deviation | 113.162 | 5 | 22.632 | - | - |
| Total | 10,198.197 | 12 | - | - | - |
| Correction | 3335.883 | 11 | - | - | - |

A small amount of polypropylene fiber added to the foam concrete, uniformly dispersed inside the foam concrete, and this did not significantly change the compressive strength of the foam concrete. The deformation capacity of the concrete was enhanced under the action of the binding force of the fiber. When the amount of polypropylene fiber added reached more than 1.5% during the concrete mixing and casting process, the

excessive fiber was closely distributed inside the foam concrete, the gap between the fibers increased, and air was mixed into the gap between the fibers to form larger holes, which reduced the density of the concrete material and led to further weakening of the compressive strength of the foam concrete, and the longitudinal strain of the concave hexagonal unit cells decreased.

### 3.2. Deformability Analysis

The digital scattering method (DSCM) was used to compare the digital images before and after the deformation of the observation surface of the concave hexagonal unit cell, as well as to analyze and obtain the deformation information of the observation surface of the object. This was conducted by matching the calculated sub-regions before and after the deformation, based on the correlation search calculation of the gray value distribution characteristics. The displacement field was determined by measuring the displacement components u and v of the sample points $(x_0, y_0)$ on the surface of the concave hexagonal cell in an axial two-dimensional way, and the lateral displacement nephogram of the concave hexagonal cell during static compression was obtained. In this experiment, the deformation when the specimen was damaged was taken as the equivalent deformation, and the displacement nephogram of concave hexagonal cells with different fiber contents under peak stress was made, as shown in Figure 4.

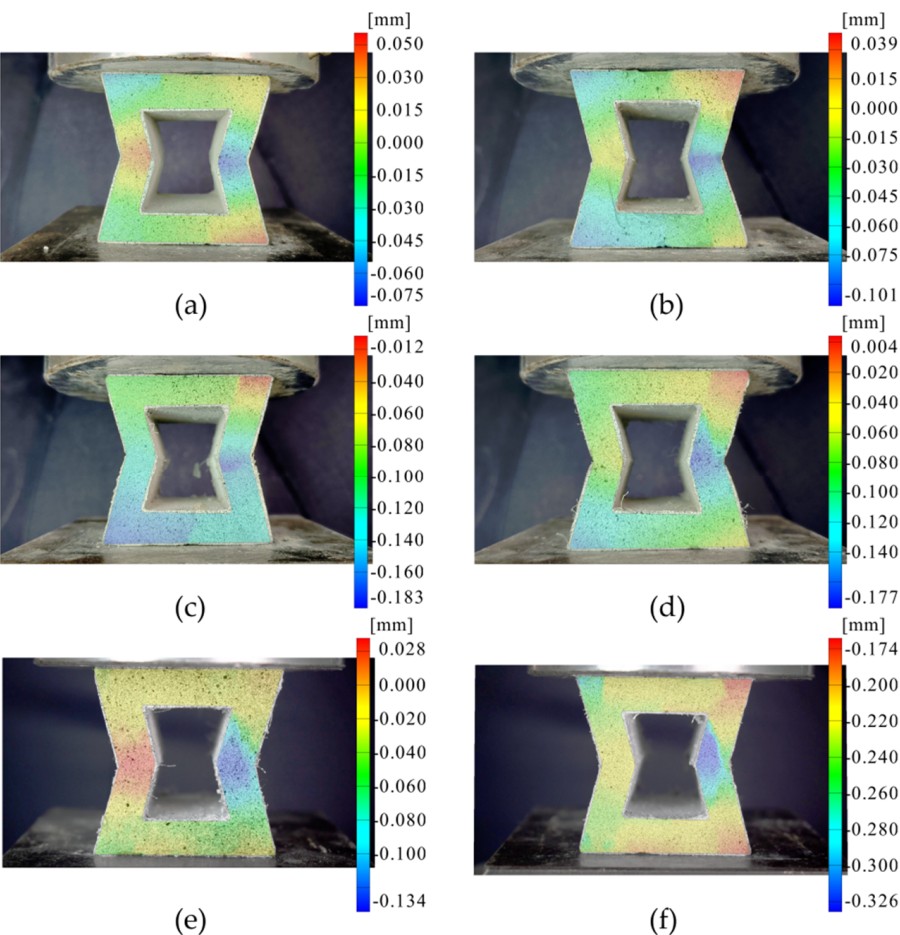

**Figure 4.** Cloud diagram of lateral displacement of the concave hexagonal unit cell: (**a**) A-0; (**b**) A-0.5; (**c**) A-1.0; (**d**) A-1.5; (**e**) A-2.0; (**f**) A-2.5.

Figure 4 shows that during the deformation of the concave hexagonal unit cells with different fiber contents, the displacement at the middle of the concave part on both sides of the cell wall is the largest, and its displacement direction is inside the concave hexagonal

unit cell. With the increase in fiber content, the displacement change demarcation line appears more and more obvious at the side of the cell wall, as shown in Figure 4d–f. The displacement of the material changes sharply at this place, which indicates that the foam concrete material has a larger deformation at this place before the structural damage of the concave hexagonal unit cells. Compared with the two sides of the cell wall, the transverse displacement changes below and above the cell wall are smaller, and by comparing the displacement distribution at the two ends of the cell wall, the displacement at the left end is smaller than the displacement at the right end, which indicates that the tensile deformation occurs both below and above during the compression process. Therefore, the deformation pattern of the concave hexagonal unit cells during static compression is as follows: the upper and lower part of the cell wall stretches outward, while the concave parts on the left and right sides bend and deform under pressure, and the middle part shrinks inward.

The width of the transverse displacement interval is the maximum difference of transverse displacement of the material, which can visually reflect the ability of transverse deformation of the material. According to the transverse displacement cloud diagram of the concave hexagonal unit cells, the transverse displacement interval width of the samples were compared. The transverse displacement interval width of the sample with 0% fiber content was 0.135 mm, while the transverse displacement interval values of the other five groups of concave hexagonal unit cells were 0.140 mm, 0.172 mm, 0.181 mm, 0.162 mm, and 0.152 mm in order, which proves that the variation of the content of polypropylene fibers can affect the transverse deformation ability of the concave hexagonal unit cell. In Figure 4d, it can be observed that the transverse displacement variation of the concave hexagonal unit cells with 1.5% fiber content was greater than that with other fiber contents, and the width of its transverse displacement interval increased by 34.1% compared with that of the concave hexagonal unit cells with 0% fiber content.

Poisson's ratio is an elastic constant reflecting the transverse deformation of materials. Since the sample structure used in this test was the concave hexagonal cell, sampling points were set in the concave portions of both sides of the cell wall to obtain the variation of the transverse distance between points. The measured longitudinal and transverse deformation is shown in Figure 5.

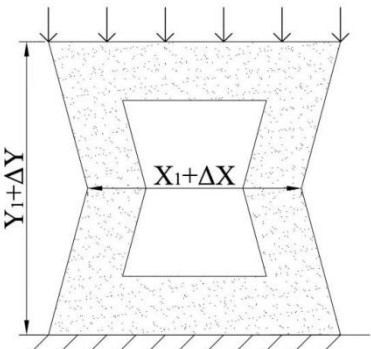

**Figure 5.** Schematic diagram of deformation measurement of the concave hexagonal unit cell.

Taking the sample A-0-1 as an example, the transverse deformation data were obtained from the images of the compression process by the optical experiment software GOM Correlate, the curves were compiled as shown in Figure 6a, and the transverse deformation of the specimen at the time of damage was used as the equivalent transverse deformation $\Delta X$. Figure 6b shows the curve compiled from the longitudinal deformation data recorded by the RMT system in the static compression process. The longitudinal deformation of the specimen at the time of damage was used as the equivalent transverse deformation $\Delta Y$.

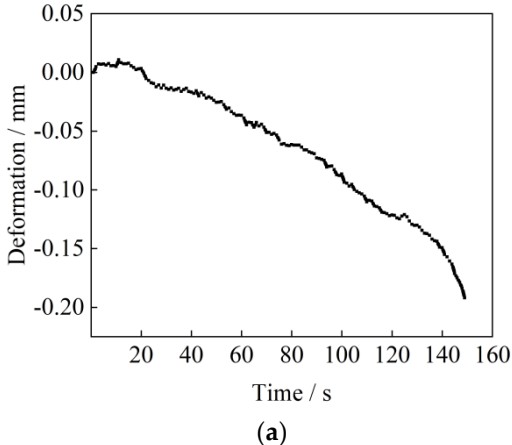 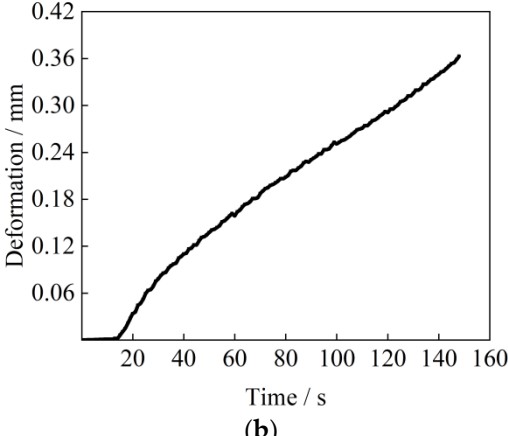

**Figure 6.** Transverse deformation curve of the concave hexagonal unit cell (A-0-1): (**a**) transverse deformation ΔX; (**b**) longitudinal deformation ΔY.

Due to longitudinal strain, $\varepsilon_y$ can be directly obtained from the recorded data of the RMT system during the static compression of the specimen, so the Poisson's ratio v of the concave hexagonal unit cell is calculated as follows:

$$\nu = -\frac{\Delta X}{X_0} \cdot \frac{Y_0}{\Delta Y} = -\frac{\Delta X}{X_0} \cdot \frac{1}{\varepsilon_y}, \tag{1}$$

where $\varepsilon_y$ is the longitudinal strain of the concave hexagonal unit cell. Table 4 shows the Poisson's ratios of the samples of the concave hexagonal unit cell with different fiber contents, calculated from the strain data of the experimentally obtained specimens.

**Table 4.** Poisson's ratios of different concave hexagonal cells.

| Sample | A-0 | A-0.5 | A-1.0 | A-1.5 | A-2.0 | A-2.5 |
|---|---|---|---|---|---|---|
| 1 | −0.072 | −0.101 | −0.131 | −0.149 | −0.132 | −0.105 |
| 2 | −0.068 | −0.095 | −0.138 | −0.142 | −0.129 | −0.109 |
| 3 | −0.076 | −0.099 | −0.136 | −0.140 | −0.130 | −0.106 |

The trend of the effect of fiber content variation on Poisson's ratio of concave hexagonal unit cells obtained by finishing is shown in Figure 7. The negative Poisson's ratio of the concave hexagonal unit cells with 0–1.5% fiber volume content increased with the increase in fiber content; the increase in fiber content over 1.5%, on the contrary, decreased the negative Poisson's ratio of concave hexagonal unit cells.

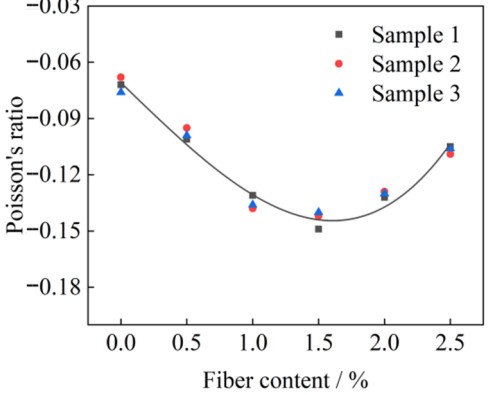

**Figure 7.** Effect of fiber content on Poisson's ratio of concave hexagonal unit cells.

It can be seen from Figure 7 that the addition of polypropylene fiber can reduce the Poisson's ratio of concave hexagonal cells. When the fiber content is 0–1.0%, the foam concrete has a better constraint effect, which makes the strain value of foam concrete in static compression increase, thus reducing the Poisson's ratio of concave hexagonal cells. When the volume of polypropylene fiber is increased to 1.0–1.5%, the compressive strength of foam concrete increases, and the fiber also has a good restraining effect on the material. With the increase in fiber content, the deformation at the concave point of both sides of the sample increases, thus effectively reducing the Poisson's ratio of the concave hexagonal unit cell. When the volume content of polypropylene fibers exceeds 1.5%, the strength of foam concrete weakens substantially, and its deformation ability also becomes weaker, so the Poisson's ratio of the concave hexagonal unit cell increases.

In summary, the appropriate amount of polypropylene fibers can effectively increase the deformation capacity of the concave hexagonal unit cells of foam concrete, and the Poisson's ratio of the concave hexagonal unit cells of foam concrete with 1.5% polypropylene fibers is the smallest.

### 3.3. Destruction Process

In the static compression test of the concave hexagonal unit cells, a difference in the process of destruction of the concave hexagonal unit cells with different fiber contents could be observed. Three types of hexagonal unit cells with different fiber contents were classified as no fiber addition, low fiber addition, and high fiber addition. Figure 8 shows the damage process of three different types of concave hexagonal unit cells in the static compression test.

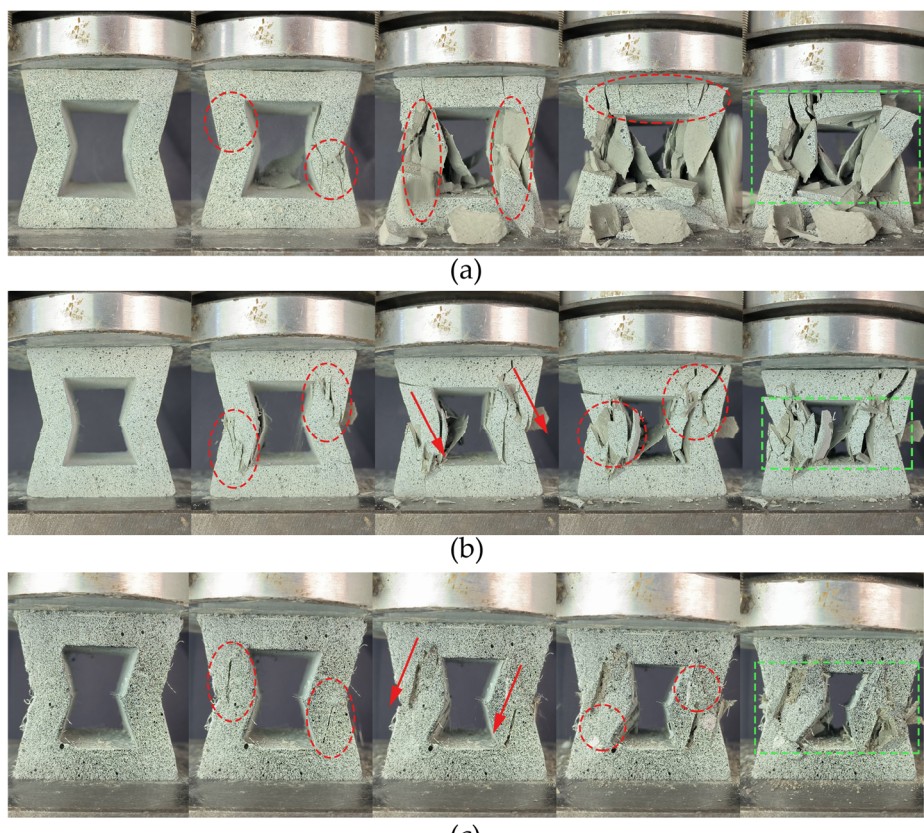

(a)

(b)

(c)

**Figure 8.** Failure process of different concave hexagonal cell specimens: (**a**) A-0; (**b**) A-0.5; (**c**) A-2.5.

For the concave hexagonal unit cell with 0% fiber content, the damage process is shown in Figure 8a. During the compression process, the cell wall first showed an oblique crack on the left and right sides along the concave edge of the hexagon, basically in the

form of "upper left and lower right" or "lower left and upper right" distribution. With the increase in pressure, multiple secondary cracks rapidly extended around the oblique cracks and extended to the upper and lower pressure-bearing surfaces of the cell wall. The cracks penetrated each other to cause the cell wall to break up, and there were pieces of debris spalling down continuously. The specimen was completely destroyed; the concave parts of the left and right sides of the cell wall were more thoroughly damaged, and the supporting surface was also damaged by crack penetration. Since the damage process of the specimen developed rapidly and the fracture surface was rough, it can be judged that the damage process of the unit cell specimen with 0% fiber content belonged to the category of brittle damage.

For concave hexagonal unit cells containing a small amount of polypropylene fiber, a sample with 0.5% fiber content was taken as an example, and its failure process is shown in Figure 8b. Under pressure, oblique cracks appeared on the left and right sides of the cell wall in a "upper left and lower right" or "lower left and upper right" distribution, and the oblique cracks continued to increase under load, resulting in the destruction of the cell wall structure through the front and back of the specimen. The oblique crack through the specimen was divided into upper and lower parts. The upper part slid downward along the oblique crack under the continuously increasing load, and extruded the lower part. The concave surfaces in the cell wall produce secondary cracks during the extrusion process, causing the part to continue to crack. After the compression of the specimen, it could be observed that the concave parts on the left and right sides of the cell wall broke into several pieces, while there were only a few cracks distributed on the pressure-bearing surface. Although the specimen maintained good integrity after the damage, the connection between the fragments was not tight.

With the increase in fiber content, the effect of fiber on concrete material was more obvious. For concave hexagonal unit cells with high polypropylene fiber content, a sample with 2.5% fiber content was taken as an example, and its failure process is shown in Figure 8c. The specimen was penetrated by oblique cracks distributed on the left and right sides of the cell wall, showing "upper left and lower right" or "lower left and upper right" distribution, and was then divided into upper and lower parts. The upper part slowly slid downward along the oblique crack under the action of a continuously increasing load, and squeezed the lower part. In this process, a few secondary cracks were generated around the oblique crack. After the compression of the specimen, the concave portions on the left and right sides of the cell wall broke into fewer pieces. There were more concrete particles at the crack, and there was no obvious crack on the pressure-bearing surface of the cell wall. Concave hexagonal unit cells with high polypropylene fiber content had better integrity after destruction, and the connection between fragments was tighter than in cells with low polypropylene fiber content.

In the process of compression failure of the concave hexagonal unit cell specimen, firstly, oblique cracks with the distribution of "upper left and lower right" or "lower left and upper right" were generated along the concave edge of the concave hexagon on the left and right sides of the cell wall. The oblique cracks generated secondary cracks under the load, and then the secondary cracks continued to extend to the periphery, and finally led to the destruction of the specimen. In this process, the energy of the concave hexagonal unit cell was transformed into different forms, including the rupture energy of the foam cell wall, the bending deformation energy of the cell wall, and the fracture energy of the foam concrete. The degree of damage to the left and right sides of the cell wall was relatively high, while the degree of damage to the upper and lower bearing surfaces was relatively small, indicating that the left and right concave parts of the cell wall played a major role in energy absorption during the ballasting process. Adding fiber can give the specimen better integrity after compression failure, and part of the energy is converted into the fracture energy of fiber during compression. With the fiber content increasing, the failure process of the specimen becomes relatively slow, and the number of cracks also decreases.

### 3.4. Energy Absorption Capacity

The influence of different fiber contents on the peak stress and strain of the concave hexagonal unit cell of foam concrete was obtained through the previous analysis. Figure 9 shows the stress-strain relationship curve in the static compression experiment of the concave hexagonal unit cell samples.

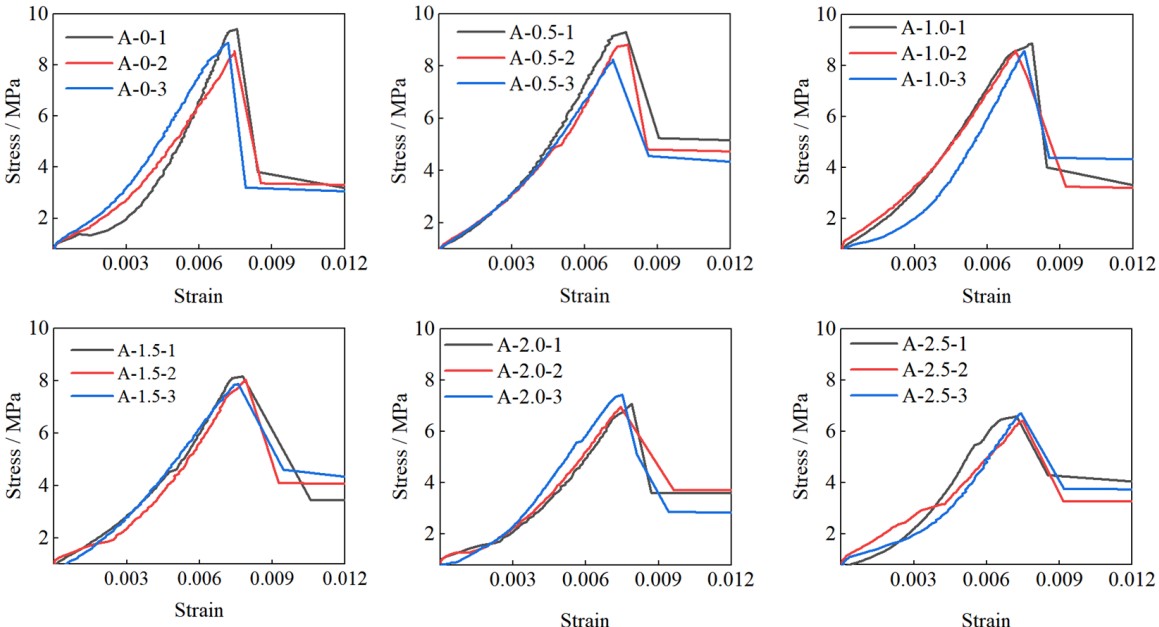

**Figure 9.** Stress-strain relationship curve of concave hexagonal unit cell specimen.

The energy absorption efficiency is used to evaluate the energy absorption capacity of the test specimen. The commonly used parameters are energy absorption efficiency, energy absorption capacity of foam concrete per unit volume $w$, total energy absorption $E_a$, specific energy absorption $E_{sa}$, etc.

$$E_f(\varepsilon_a) = \frac{1}{\sigma_a} \int_0^{\varepsilon_D} \sigma(\varepsilon)d\varepsilon, 0 \le \varepsilon \le 1, \tag{2}$$

where $\varepsilon_a$ is the strain value of the specimen at a certain moment, $\sigma_a$ is the stress value corresponding to it, and $\varepsilon_D$ is the compaction strain, i.e., the strain corresponding to the maximum value of energy absorption efficiency. In practical application, it is necessary to combine the actual foam concrete stress-strain curve to determine the compaction strain of the specimen. The energy absorption capacity $w$ per unit volume of foam concrete is defined as:

$$w = \int_0^{\varepsilon_D} \sigma(\varepsilon)d\varepsilon, \tag{3}$$

The total energy $E_a$ absorbed by the foam concrete specimen was obtained by calculating the area enclosed by the load-displacement curve of the specimen, the significance of which is that the greater the total energy absorbed, the better the impact resistance of the material:

$$E_a = \int_0^l Pdl, \tag{4}$$

where $l$ is the displacement value corresponding to $\varepsilon_D$, and $p$ is the magnitude of the load corresponding to the displacement of l. The ratio of the total energy absorbed by the test block to its mass, i.e., the specific absorbed energy $E_{sa}$, is calculated by:

$$E_{sa} = \frac{E_a}{m}, \tag{5}$$

Since, in this experiment, the specimen was judged to be damaged when it was sharply deformed, the energy absorbed by the specimen compressed to the peak strain was used here as a judgment index to judge the energy absorption capacity of the specimen. According to Equation (4), the total energy of the hexagonal specimen during compression deformation was obtained, and the specific energy absorption was calculated by using Equation (4) in combination with the mass of the sample, weighed before the experiment. The masses and energy absorption indexes of different hexagonal unit cells are shown in Table 5.

**Table 5.** Mass and energy absorption index of different concave hexagonal unit cells.

| Sample | Quality/g | Average/g | Ea/J | Average/J | Esa/J·kg$^{-1}$ | Average/J·kg$^{-1}$ |
|--------|-----------|-----------|------|-----------|-----------------|---------------------|
| A-0-1 | 133.6 | | 3.599 | | 26.943 | |
| A-0-2 | 135.4 | 135.1 | 3.664 | 3.736 | 27.060 | 27.644 |
| A-0-3 | 136.4 | | 3.946 | | 28.930 | |
| A-0.5-1 | 135.3 | | 4.414 | | 32.623 | |
| A-0.5-2 | 133.2 | 134.3 | 4.226 | 4.221 | 31.727 | 31.431 |
| A-0.5-3 | 134.4 | | 4.024 | | 29.943 | |
| A-1.0-1 | 132.2 | | 4.313 | | 32.625 | |
| A-1.0-2 | 132.6 | 131.8 | 3.743 | 3.881 | 28.230 | 29.440 |
| A-1.0-3 | 130.6 | | 3.587 | | 27.464 | |
| A-1.5-1 | 131.6 | | 3.323 | | 25.253 | |
| A-1.5-2 | 130.2 | 130.4 | 3.731 | 3.580 | 28.654 | 27.471 |
| A-1.5-3 | 129.3 | | 3.686 | | 28.504 | |
| A-2.0-1 | 126.3 | | 3.283 | | 25.994 | |
| A-2.0-2 | 124.5 | 124.5 | 2.992 | 3.191 | 24.0330 | 25.629 |
| A-2.0-3 | 122.8 | | 3.298 | | 26.860 | |
| A-2.5-1 | 121.5 | | 2.961 | | 24.374 | |
| A-2.5-2 | 120.7 | 120.8 | 3.105 | 2.937 | 25.723 | 24.307 |
| A-2.5-3 | 120.3 | | 2.746 | | 22.826 | |

The total energy absorption and specific energy absorption of the concave hexagonal unit cell with 0.5% fiber content are larger than those with other fiber content. In static compression tests, the total energy absorbed by concave hexagonal cells with 0.5% fiber content increased by 12.98% compared to concave hexagonal cells without fibers, indicating that a small amount of polypropylene fibers (about 0.5%) contributed to the higher amount of energy absorbed by concave hexagonal cells. Figure 10 was obtained by collating the average values of total and specific absorbed energy for each group of specimens, demonstrating that when the polypropylene fiber content exceeds 0.5%, the total and specific absorbed energy of the concave hexagonal unit cells decreases with the increase in fiber content.

The concave hexagonal unit cell mainly transforms energy into the rupture energy of the foam pore wall and the bending deformation energy of the cell wall in the process of structural deformation. During the static ballast process, the concave portion of the cell wall of the hexagonal specimen has the main role in energy absorption. As the load strength increases, the specimen deforms, and the concave portion of the cell wall shrinks inward, deforms, and bends, which makes the foam concrete subject to force. The addition of polypropylene fiber can effectively provide an additional binding force and increase the tensile strength and enhance the material toughness. When the polypropylene fiber content is 0–1.5%, it can not only enhance the material toughness of the concrete, but also increase the porosity of the concrete. However, the impact on the compressive strength of the concrete is not significant. The energy absorbed by the deformation and fracture of the foam pore wall in the compression process is increased. At this point, the specific energy absorption of the specimen is enhanced relative to that of the sample without fibers.

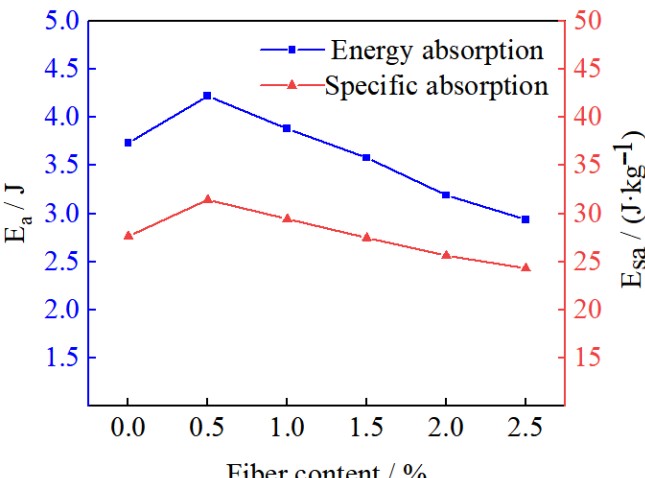

**Figure 10.** Effect of fiber content on energy absorption capacity of the concave hexagonal unit cell.

Figure 11 shows the microscopic morphology of cracks in the concave hexagonal unit cell at low (0.5%) and high (2.5%) fiber content, respectively. The exposed polypropylene fibers at the high fiber content crack in Figure 11b are longer than those at the low fiber content crack in Figure 11a, which indicates that the polypropylene fibers break free from the foam concrete under force when the specimen is structurally damaged. With the increase in polypropylene fiber content, the number of pores in the foam concrete also increases, which leads to the thinning of the internal pore walls and a reduction in the contact surface between the polypropylene fibers and the concrete material. The increase in porosity leads to a reduction in the peak stress of the specimens and the inability of the polypropylene fibers to exhibit the inhibiting effect during the compression of the specimens. In higher stress conditions, polypropylene fibers have not yet played a restraining role; the foam concrete begins to rupture in advance, so the energy absorbed by the deformation and fracture of foam concrete is reduced.

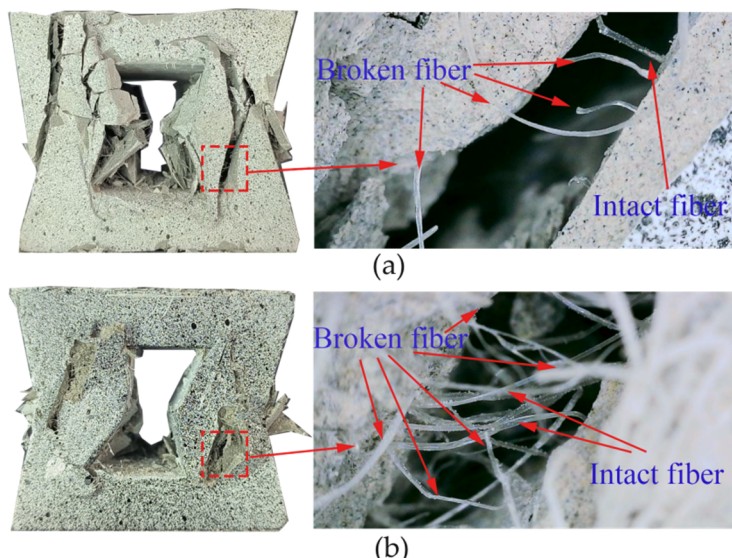

**Figure 11.** Microscopic morphology of cracks in the concave hexagonal unit cell: (**a**) low fiber content; (**b**) high fiber content.

A small amount of polypropylene fiber content can effectively increase the energy absorption efficiency of the concave hexagonal unit cell. In the process of static compression of foam concrete, the internal pore wall will continuously produce brittle fracture damage,

thus exerting the energy absorption ability of the material. The addition of polypropylene fibers can provide a binding force for the internal pore walls and convert part of the energy into fracture energy of polypropylene fibers after structural damage, thus increasing the energy absorption of the concave hexagonal unit cell.

## 4. Conclusions

In this paper, a kind of concave hexagonal unit cell of polypropylene fiber foam concrete is proposed. Through a quasi-static compression test, the concave hexagonal unit cells of foam concrete with different fiber contents are studied, and the influence of fiber content on the mechanical properties of concave hexagonal unit cells is analyzed.

- The increase in polypropylene fiber volume content reduces the compressive strength of foam concrete under static compression, but the peak stress of the concave hexagonal unit cell decreases less rapidly than that of the cube specimen with the same size.
- Adding the proper amount of polypropylene fiber into the concave hexagonal cell structure of foam concrete can improve the toughness and reduce the Poisson's ratio of concave hexagonal cells. The effect is best when the fiber content is 1.5%, the width of its transverse displacement interval is increased by 34.1% compared with that of the concave hexagonal unit cells with 0% fiber content, and it has the lowest Poisson's ratio.
- In the process of static compression, the concave hexagonal unit cell is the first to crack at the concave portion of the cell wall of the specimens, and the damage is the most thorough there, indicating that the left and right concave surfaces of the cell wall play a major role in energy absorption during the process of ballasting. Additionally, the cracks are distributed in the form of "upper left and lower right" or "lower left and upper right".
- Adding the proper amount of polypropylene fiber can significantly improve the energy absorption efficiency of concave hexagonal cells. When the content of polypropylene fiber is 0.5%, the total energy absorption of concave hexagonal cells increases by 12.98%. When the fiber content exceeds 0.5%, excessive polypropylene fiber reduces the strength and deformation of the specimen greatly, thus reducing the energy absorption efficiency.

**Author Contributions:** Conceptualization, Z.S. and Z.Y.; methodology, Z.Y.; software, L.G.; validation, H.W., J.W. and C.Q.; formal analysis, Z.S.; investigation, Z.S.; resources, Z.S.; data curation, Z.S.; writing—original draft preparation, Z.S.; writing—review and editing, Z.Y.; visualization, H.W.; supervision, Z.Y.; project administration, Z.Y.; funding acquisition, Z.Y. All authors have read and agreed to the published version of the manuscript.

**Funding:** This research was supported by Zhiqiang Yin, the funding number were No. 52274070, 51874006 (the National Natural Science Foundation of China) and No. 202004a07020045 (Anhui Provincial Key Research and Development Program).

**Institutional Review Board Statement:** Not applicable.

**Informed Consent Statement:** Not applicable.

**Data Availability Statement:** The experimental test data used to support the findings of this study are available from the corresponding author upon request.

**Conflicts of Interest:** The authors declare no conflict of interest. The funders had no role in the design of the study; in the collection, analyses, or interpretation of data; in the writing of the manuscript; or in the decision to publish the results.

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
