# Peer review of "Static Compressive Properties of Polypropylene Fiber Foam Concrete with Concave Hexagonal Unit Cell"

_applsci, doi:10.3390/app13010132_

Round 1

Reviewer 1 Report

General considerations.

The article is very interesting and show the new materials analyses. But I suggest that explanations about Negative Poisson's ratio can be summarized (one paragraph 5 – 7 lines).

Another important point is the balance of the article. The initial statements lead to analyses on two different samples. But the results and conclusions focus on Concave Hexagonal Unit Cell.

The results are exchange in the many times this causes the confusion during evaluation.

Reviewer 2 Report

Accept in present form

Reviewer 3 Report

In this paper, the test results of the influence of fiber on the negative Poisson's ratio effect of foam concrete are shown. A concave hexagonal unit cell structure of polypropylene fiber foam concrete are considered. The research topic of the manuscript is novel and it fits on the journal purpose.

The reviewer recommends the publication of this paper after solving the following issues:

1. In Figure 4, the stress (or force) value for each from the cloud diagrams in the DIC (Digital Image Correlation) is missing,

2. In Figure 6, the (a) Stress-Deformation and (b) Stress-Time plots are missing.

3. Could you explain so short testing time (less than 160 sec. - see Figure 6)? Usually static tests shall last about 300 (+/-120) sec.

4. Some editorial issues shall be addressed, such as:

- in rows No. 95, No. 97, No. 99: 9 mm, 50 mm x 50 mm x 50 mm with a gap between a number and units,

- in row No. 112: 50 mm x 50 mm x 50 mm shall be written in the same row, 

- in Table 1: editing of the heading row is required.

5. Also, I recommend to add some literature such as e.g. "Meskhi, B.; Beskopylny, A.N.; Stel’makh, S.A.; Shcherban’, E.M.; Mailyan, L.R.; Beskopylny, N.; Chernil’nik, A.; El’shaeva, D. Insulation Foam Concrete Nanomodified with Microsilica and Reinforced with Polypropylene Fiber for the Improvement of Characteristics. Polymers 2022, 14, 4401. https://doi.org/10.3390/polym14204401"
